# Assessment of Sustainable Collaboration in Collaborative Business Ecosystems †

Paula Graça [1,2] and Luis M. Camarinha-Matos [1,*]

1 School of Sciences and Technology and Uninova CTS, NOVA University of Lisbon, Campus de Caparica, 2829-516 Caparica, Portugal; paula.graca@isel.pt

2 Instituto Superior de Engenharia de Lisboa, Instituto Politécnico de Lisboa, Rua Conselheiro Emídio Navarro 1, 1959-007 Lisbon, Portugal

* Correspondence: cam@uninova.pt

† This paper is an extended version of the paper entitled AI and Simulation for Performance Assessment in Collaborative Business Ecosystems: Graça P., Camarinha-Matos L.M. (2021) AI and Simulation for Performance Assessment in Collaborative Business Ecosystems. In: Camarinha-Matos L.M., Ferreira P., Brito G. (eds) Technological Innovation for Applied AI Systems. DoCEIS 2021. IFIP Advances in Information and Communication Technology, vol 626. Springer, Cham. https://doi.org/10.1007/978-3-030-78288-7_1.

**Abstract:** Advances in information and communication technologies and, more specifically, in artificial intelligence resulted in more intelligent systems, which, in the business world, particularly in collaborative business ecosystems, can lead to a more streamlined, effective, and sustainable processes. Following the design science research method, this article presents a simulation model, which includes a performance assessment and influence mechanism to evaluate and influence the collaboration of the organisations in a business ecosystem. The establishment of adequate performance indicators to assess the organisations can act as an influencing factor of their behaviour, contributing to enhancing their performance and improving the ecosystem collaboration sustainability. As such, several scenarios are presented shaping the simulation model with actual data gathered from three IT industry organisations running in the same business ecosystem, assessed by a set of proposed performance indicators. The resulting outcomes show that the collaboration can be measured, and the organisations' behaviour can be influenced by varying the weights of the performance indicators adopted by the CBE manager.

**Keywords:** collaborative networks; business ecosystem; performance indicators; simulation; agent-based modelling



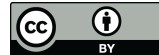

## 1. Introduction

Advances in information and communication technologies and, more specifically, in artificial intelligence (AI) have enabled more intelligent systems, which, in the business world, particularly in collaborative business ecosystems (CBEs), can streamline collaborative processes among organisations, promoting sustainability and resilience [1]. In fact, collaboration is increasingly recognised as a critical component of the industry efforts to address sustainability challenges [2].

The notion of CBE [3] results from a combination of Moore's view [4], who first introduced the term business ecosystem, inspired by biological ecosystems, and the collaborative networks (CN) view, as presented in Camarinha-Matos and Afsarmanesh's taxonomy of a CN [5,6], to highlight the collaboration facets of our view of a business ecosystem. Collaborative networks, when constituted by appropriate partners, will encourage knowledge and information sharing to strengthen products, processes, and market innovation, resulting in improved business performance [7]. An essential aspect in this context is the evaluation of the collaboration performance among organisations in the ecosystem to identify potential earnings and promote sustainability of the collaboration.

Inter-organisational collaboration has gained increased attention in research, given its documented influence on the innovation processes of small and medium-sized enterprises (SMEs) [8]. Indeed, many studies can be found in the literature at the network level, supporting broad theories on inter-organisational networks, many of them highlighting their associated capabilities [9]. However, only a few studies on business networks can be found particularly addressing the evaluation of their collaboration [10], despite some performance assessment attempts. Some examples of such attempts include:

- Proposals of performance indicators for CNs based on collaboration benefits, as well as indicators for relationships and assets analysis [11–13];
- A number of different approaches to assess collaboration in CNs, namely a model for evaluating collaboration attributes in cluster-based companies [14], a study to examine CNs' effect on SMEs' business performance [7], and a method for the measurement of the social dimension of (cognitive) trust factors in CNs [15];
- Suggestions of using balanced scorecards and key performance indicators in CNs [16–18] and supply chain management [19–21];
- Various performance measures, metrics, and methods for supply chain and supply chain collaboration (SCC) [22–26], although, even when it comes to SCC, there is not an appropriate measurement system by which the depth of collaboration is measured [27];
- Investigations on the applicability of social networks analysis (SNA) in inter-organisational networks of firms, namely the use of measures of network density, centrality and tie strength [28,29], Poisson regression [30], and partial least squares and fuzzy sets [31]; this further led to some contributions to the design of performance indicators based on the structural analysis of the relationships between actors in social network analysis [32].

However, despite these earlier developments, there is a need for an integrated approach to assess performance of CBEs. The identification of this need motivated our work, which is driven by the following research questions:

**RQ1.** *What is a reasonable set of performance indicators to measure and assess collaboration benefits in a CBE?*

**RQ2.** *How can performance assessment methods based on economic and social value promote sustainability in a CBE?*

In order to respond to these research questions, this article presents a case study using a Performance Assessment and Adjustment Model (PAAM), previously proposed in [33]. The PAAM is a simulation model representing a business environment populated with the agents representing the organisations whose behaviour is modelled using actual data from the business activity (the year 2019) of companies in the information technology (IT) industry running in the same ecosystem. The organisations included in the CBE have different collaborative behaviours, classified into classes of collaboration willingness that define their profile. Consequently, it is assumed that, when subjected to a performance assessment, as proposed in [34], they respond differently, similar to individuals. In other words, when organisations know how they are "measured", they are expected to adjust their behaviour to increase their results. Thus, this case study aims to show that organisations' collaboration performance and that of the ecosystem can be improved using an influence mechanism, whereby the CBE manager can vary the weights associated with the adopted performance indicators, causing a percentage of variation by a given influencing factor on the organisations' behaviour, expecting to increase their performance in the desired direction.

The remaining part of the article, which is an extended version of a conference paper [35], presenting the research design, new experiments, and simulation results, is organised as follows: Section 2 describes the research design that guided this work, encompassing it into the three cycles of design science research [36]; Section 3 explains the designed artefacts, the simulation model including the performance assessment and influence mechanism; Section 4 details the internal collaborative behaviour of an agent; Section 5 presents and discusses the results of different scenarios of simulation using the

actual data collected from companies in the IT industry; finally, the last section contains a summary of the research contributions, refers main limitations and identifies future work.

## 2. Research Design

The field of information systems aims to improve knowledge on the application of ITs. It purposely designs human–machine artefacts that significantly impact people, organisations, and society [36]. In this domain exists two complementary research paradigms: behavioural science and design science. The behavioural science paradigm "*seeks to discover and verify laws or principles that explain or predict human or social behaviour*". In contrast, the design science paradigm "*seeks to extend the boundaries of human beings and social capabilities by creating new and innovative artefacts*", namely developing technology-based solutions to important and relevant business problems [36].

Taking into account our research goals of designing novel artefacts, we adopted the design science research (DSR) paradigm according to Hevner et al. [36], although there are other variants, such as the example in Peffers et al. [37]. The novel artefacts are a *Performance Assessment*, made of a set of performance indicators adequate to assess collaboration of the organisations in a CBE, and a simulation model, the PAAM, to support the performance indicators' evaluation, which is expected to act as a factor of influence of the organisations, according to an Influence Mechanism, to improve their behaviour.

According to the adopted DSR [36,38], the research process is viewed as an embodiment of three closely related cycles of activities that must be present and clearly identifiable [38]:

- Relevance Cycle: connects the contextual environment of the research project and the design science activities;
- Design Cycle: iterates between the core activities of building and evaluating the designed artefacts of the research project;
- Rigor Cycle: connects design science activities with the knowledge base of scientific foundations that inform the research project.

The following sections extend the description and meaning of each cycle in the context of this work, described in Figure 1.

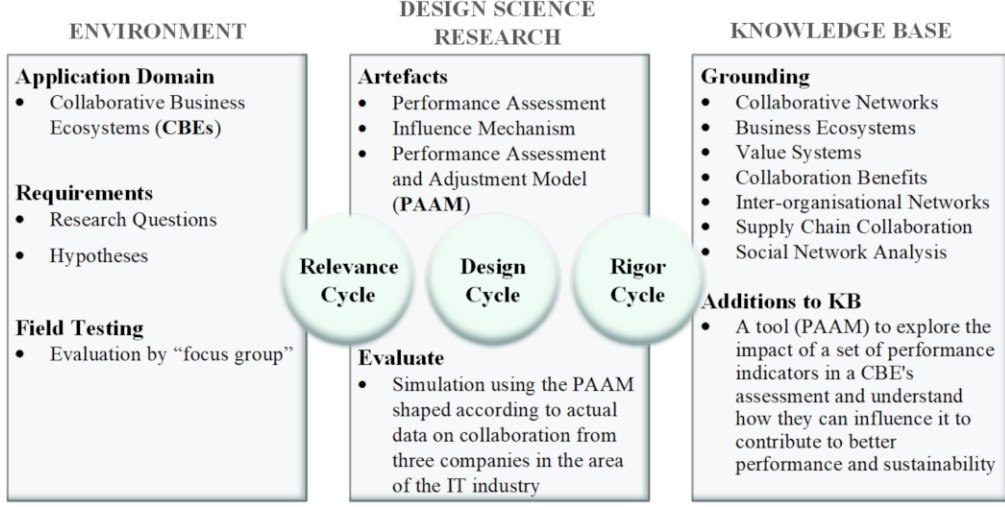

**Figure 1.** Design science research process followed by this dissertation (adapted from Hevner et al. [36], Hevner [38], and Hevner and Chatterjee [39]).

### 2.1. Requirements

The relevance cycle initiates with the opportunity/problem to be addressed in the application domain as input and the acceptance criteria for evaluating the research results [38]. The application domain of this research is a collaborative business ecosystem, and the problem is identified and expressed by the research questions **RQ1** and **RQ2** stated in the previous section.

The following corresponding hypotheses guide our research to find answers for the questions:

**Hypothesis 1.** *Collaboration benefits can be evaluated and made explicit if a set of indicators is established through a holistic combination of value and benefit concepts derived from a number of research areas such as value systems, collaboration benefits, inter-organisational networks, supply chain collaboration, and social networks analysis.*

**Hypothesis 2.** *Performance indicators are a useful mechanism for assessing a CBE if they can contribute as a factor of influence for organisations to evolve and self-adjusting their behaviour, thereby improving the ecosystem performance and sustainability.*

It is broadly accepted that collaboration benefits the actors involved, allowing divergent thinking to develop new understandings [40], facilitating innovation and services, and reducing or eliminating conflicts [41]. The literature on CNs presents significant evidence of potential benefits of collaboration, namely works on benefits analysis [12] and value systems for sustainable collaboration [42].

*2.2. Grounding*

The foundations for the artefacts of the research design consider contributions based on various literature findings, as illustrated in Figure 2. A number of research areas were considered as a basis for this work, including collaborative networks, business ecosystems, value systems, collaboration benefits, inter-organisational networks, supply chain collaboration and social networks analysis.

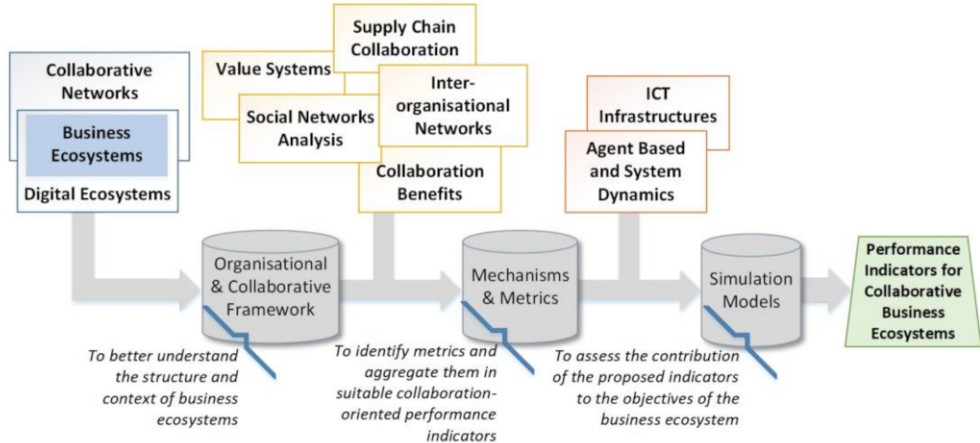

**Figure 2.** Foundations for the design of artefacts.

The area of collaborative networks offers a CN taxonomy locating a business ecosystem and provides a reference model for CNs [5,43,44]. This reference model allows a structural and behavioural definition of a CBE. On the other hand, the efforts made in digital ecosystems, though with more emphasis on computational models, allowed the characterisation of a business ecosystem through a set of key attributes derived from what the author considered a generic ecosystem [45,46]. The combination of these two lines of research contributed to the development of an organisational and collaborative framework to understand the structure, behaviour, and dynamics of a CBE.

The other mentioned areas inspired the metrics and mechanisms for a CBE in the following context:

- The value systems area identifies collaborative economic and social core values and provides mechanisms to access the network participants' value systems' alignment, thereby allowing detection of potential conflicts affecting the network's perfor-

mance [47,48]. These mechanisms highlight important metrics when applied to a CBE for performance measurement;

- The collaboration benefits area identifies and suggests a set of performance indicators to assess benefits resulting from the collaboration in CNs [11–13]. These benefits are not particularly tailored to CBEs but can be used as a basis to a better understand collaboration benefits in a business environment;
- The supply chain collaboration area identifies collaboration to improve performance in traditional supply chains (SC) and proposes a wide diversity of methods, metrics, and mechanisms from which [22–27,49–52] are relevant examples among many others. These contributions, although focusing on a different class of CNs, can be considered as a valuable input when it comes to establishing performance indicators for CBEs;
- The inter-organisational networks area associates the structure of a network of organisations with social capital, performance, and power [9,10,53,54]. It also explains the network influence and evolution [54,55], constituting a valuable contribution to design the influence mechanism of the PAAM;
- The social networks analysis area provides a foundation for analysing and understanding social and economic networks [32,56,57], in which metrics of centrality are tailored for the performance assessment of the PAAM.

### 2.2.1. Organisational and Collaborative Framework

A general framework proposed and validated in [58] characterises the collaboration performance in SCs. It is composed of a collaboration characterisation model and a collaboration-oriented performance model. The first one represents the collaborative situation of a company and leads to the construction of its collaborative profile [58]. Following this line of research, we propose a framework to understand and model a CBE, described in terms of its structural organisation and collaborative behaviour, as depicted in Table 1. A CBE model is represented as a business environment composed of organisations collaborating in response to market opportunities. These collaborations are relationships between the organisations expressed in collaboration opportunities (*CoOps*), represented by links, which, in turn, have a weight that corresponds to the number of times the organisations collaborate (*#CoOps*).

Organisations have different profiles classified into Classes of Collaboration Willingness, which characterise their collaborative behaviour when realising market opportunities. In the implemented artefact, each profile is composed of a set of attributes, namely *contact rate*, *accept rate*, and *new products rate*, whose decimal values between 0 and 1 express a collaboration intensity factor, i.e., the propensity to collaborate by sending invites to other organisations, the inclination to accept the invitations, and in particular, those that are associated with innovation.

It is possible to instantiate any number of organisations of different profiles to set up a CBE population. The profiles are characterised by classes of collaboration willingness generically designated by Class A, B, C, . . . , Z.

### 2.2.2. Mechanisms and Metrics

Considering the benefits of collaboration, we are interested in identifying metrics that make it possible to assess them for business ecosystems. Based on the literature review, we concluded that a combination of SNA's measures of density and centrality [32,57], CNs' metrics and indicators [11–13], and findings in inter-organisational networks [9,10,53,54,59] can result in a suitable approach as the main source of inspiration.

The study of SNA applied to inter-organisational networks has been the subject of increasing research in recent years. The structure of the network formed by organisations, considering the ties between them and their strength, has a significant influence on their behaviour and performance [9]. Network structure denotes a social capital metaphor, which means a competitive advantage due to the location of individuals or groups in the social structure [53]. Structural holes (weaker links between groups) create opportunities

for those who cross the holes due to greater access to information and communication. On the other hand, networks with closure (a dense network where everybody is connected) can be essential to realising the value buried in the holes [53].

According to the literature review, predominant research at the network level in inter-organisational networks is focused on the characteristics of the whole network, such as density, centrality and cliques [10]. Findings on the nature of the ties between two organisations suggest that strong links increase trust, lowering transaction costs and increasing benefits [9]. Finally, the organisation level findings state that a high degree of centrality is positively related to their performance and that structural holes and closure generate social capital [9].

Following the main lines of research and findings exposed above, we propose a set of collaboration metrics for business ecosystems, described in Table 2. The metrics include the number of organisations in the CBE, the number of collaboration opportunities created, the weighted degree and betweenness centrality of the organisations, the number of virtual organisations created, and the number of products, services, and patents generated with or without innovation.

**Table 1.** Framework to model the structure and collaborative behaviour of a CBE.

| Description of a CBE Model |
| --- |
| A CBE is a network of organisations, connected by relationships that mean the market opportunities they share collaborating, called collaboration opportunities, to accomplish business opportunities. |

| **Structural Organisation** | |
| --- | --- |
| **Name** | **Model** |
| Collaborative Business Ecosystem (CBE) | Network of nodes |
| Organisations ($O_n$) | Nodes |
| Collaboration opportunities (CoOps) | Ties between nodes |
| Number of shared CoOps (#CoOps) | Ties' strength |

**Collaborative Behaviour**
**Classes of Collaboration Willingness**

| **Contact rate [0..1]** | **Accept rate [0..1]** | **New products rate [0..1]** |
| --- | --- | --- |
| Willingness to invite other organisations to collaborate. | Readiness to accept invitations from other organisations. | Tendency to accept opportunities related to innovation. |

**Profile of Organisations**

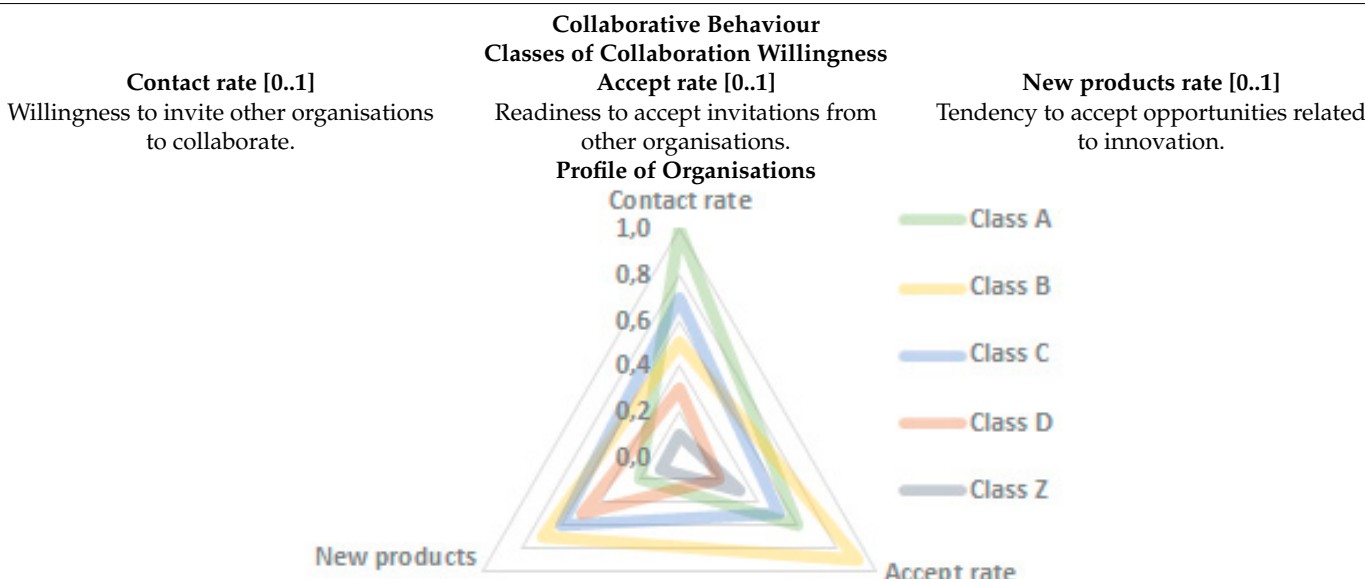

**Table 2.** Metrics to assess the organisations in a CBE individually and the CBE as a whole.

| Metric | Description |
|---|---|
| **Metrics of the Organisations $O_i \epsilon$ [$O_1$ .. $O_n$]** | |
| $O_1, \ldots, O_n$ | Organisations in the CBE |
| $\#CoOp_i$ in | No. of collaboration opportunities the organisation $O_i$ gained from the CBE |
| $\#CoOp_i$ out | No. of collaboration opportunities the organisation $O_i$ brought in the CBE |
| $\#CoOp_i$ | No. of collaboration opportunities the organisation $O_i$ participated in the CBE |
| $\#CoOp_{kj}$ | No. of collaboration opportunities between the organisation $O_k$ and $O_j$ in the CBE |
| $C_D(O_i)$ in/out | Weighted indegree/outdegree centrality ($C_D$) of the organisation $O_i$ in the CBE, which stands for the sum of direct connections in/out of $O_i$ to the n organisations $O_j$, with weight $\#CoOp_{ij}$ |
| $C_B(O_i)$ | Weighted betweenness centrality ($C_B$) of the organisation $O_i$ in the CBE, which stands for the sum of overall partial betweenness of $O_i$ relative to all pairs $O_{kj}$, assuming that connections between $O_k$ and $O_j$ have weight of $\#CoOp_{kj}$ |
| $\#VO_i$ | Number of VOs in which the organisation $O_i$ participated |
| $\#PortPd_i$ | Portfolio of products/services/patents of the organisation $O_i$ |
| $\#NewPd_i$ | Number of new products/services/patents generated by organisation $O_i$ |
| **Metrics of the CBE as a whole** | |
| $\#O$ | Number of organisations in the CBE |
| $\sum i \ \#CoOp_i$ | Total number of collaboration opportunities created in the CBE |
| $CD(O^*)$ in/out | Maximum indegree/outdegree centrality of the organisations $O_1..O_n$ |
| $CB(O^*)$ | Maximum betweenness centrality of the organisations $O_1..O_n$ |
| $\#VO$ | Number of virtual organisations created in the CBE |
| $\#PortPd$ | Total portfolio of products/services/patents of the CBE |
| $\#NewPd$ | Total of new products/services/patents generated in the CBE |

Taking into account the metrics in Table 2, we formulated a set of performance indicators described in Table 3 to assess the CBE anchored in the hypothesis **H1**, previously presented in the requirements subsection: the Contribution Indicator (CI), to measure the value creation in terms of new collaboration opportunities created in the CBE; the Prestige Indicator (PI) to measure the most prominent organisations and how collaboration among them spreads, highlighting those that are most influential; and the Innovation Indicator (II) to measure the new products, services, or patents created in collaboration.

**Table 3.** Performance Indicators to evaluate the collaboration of the organisations in a CBE individually and the CBE as a whole.

| P. Ind. | Formula | Description |
|---|---|---|
| **Performance Indicators of the Organisations $O_i \epsilon$ [$O_1$ .. $O_n$]** | | |
| **$CI_{in}$** | $CI_i in = \dfrac{C_D(O_i)in}{C_D(O^*)in} = \dfrac{\sum_j O_{ij}\#CoOp_{ij}in}{\max \sum_j O_{ij}\#CoOp_{ij}in}$ | - Assesses the contribution of organisation $O_i$ related to the number of accepted collaboration opportunities ($\#CoOp_{in}$) |
| **$CI_{out}$** | $CI_i out = \dfrac{C_D(O_i)out}{C_D(O^*)out} = \dfrac{\sum_j O_{ij}\#CoOp_{ij}out}{\max \sum_j O_{ij}\#CoOp_{ij}out}$ | - Assesses the contribution of organisation $O_i$ related to the number of created collaboration opportunities ($\#CoOp_{out}$) |
| **PI** | $PI_i = \dfrac{C_B(O_i)}{C_B(O^*)} = \dfrac{\sum_k \sum_j O_{kj}(O_i)}{\max \sum_k \sum_j O_{kj}(O_i)}$ | - Assesses the prominence/influence of organisation $O_i$ related to the number of collaboration opportunities ($\#CoOp$) |
| **II** | $II_i = \dfrac{\#NewPd_i}{\#PortPd_i}$ | - Measures the ratio of new products/services/patentes ($NewPd_i$) of the organisation $O_i$ by the total portfolio ($PortPd_i$) created |

**Table 3.** *Cont.*

| P. Ind. | Formula | Description |
|---|---|---|
| **Performance Indicators of the CBE as a whole** | | |
| **CI$_{in}$** | $CI_{CBE}in = \frac{\sum_i [C_D(O^*)in - C_D(O_i)in]}{C_D(O^*)in*(\#O-1)}$ | - Assesses the degree to which the most popular organisation [max degree centrality $C_D(O^*)$in] exceeds the contribution of the others |
| **CI$_{out}$** | $CI_{CBE}out = \frac{\sum_i [C_D(O^*)out - C_D(O_i)out]}{C_D(O^*)out*(\#O-1)}$ | - Assesses the degree to which the most active organisation [max degree centrality $C_D(O^*)$out] exceeds the contribution of the others |
| **CI** | $CI_{CBE}t = \frac{\sum_i \#CoOp_i}{\#O}$ | - Ratio of the total number of collaboration opportunities (#CoOp) created/accepted in the CBE by the total number of organisations (#O) |
| **PI** | $PI_{CBE} = \frac{\sum_i [C_B(O^*) - C_B(O_i)]}{C_B(O^*)*(\#O-1)}$ | - Assesses the degree to which the most prominent/influent organisation [max betweenness centrality $C_B(O^*)$] exceeds the contribution of the others |
| **II** | $II_{CBE} = \frac{\sum \#NewPd_i}{\sum \#PortPd_i} * r(\#VO, \#NewPd)$ | - Calculates the ratio of innovation of the organisations in the CBE, weighted by the correlation between the collaboration participation in VOs and new products/services/patents [r(#VO, #NewPd)]" |

Note: The values of the indicators are normalised between [0..1].

### 2.2.3. Simulation Models

The purpose of the performance indicators introduced in the previous subsection is to assess and influence the CBE performance and sustainability as stated in hypothesis **H2**. To this end, we built a simulation model of the CBE, whose conceptual design is depicted in Figure 3. The evaluation in DSR serves and measures how well an artefact supports a solution to the problem [37]. The selection of the evaluation method must be matched appropriately with the design artefact and the evaluation metrics [36]. Among the possible design methods to evaluate (observational, analytical, experimental and testing) [36], we chose the experimental one using a simulation model due to the lack of extensive historical concrete collaboration data from the organisations.

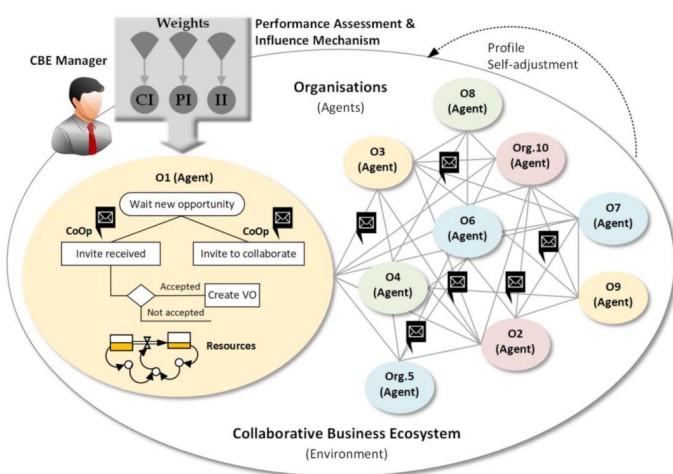

**Figure 3.** Conceptual illustration of the CBE simulation model.

A CBE model can be represented as a business environment composed of collaborative organisations that respond to market opportunities. These collaborations are relationships between the organisations expressed in collaboration opportunities (*CoOps*), represented in the model of Figure 3 by links, which, in turn, have a weight that corresponds to the number of times the organisations collaborate (*#CoOps*).

### 2.3. Artefacts

DSR provides many techniques for representing IT problems to facilitate the discovery of effective solutions [36]. The result is, by definition, an artefact, i.e., a construct, a model, a method, or an instantiation. These include, among others, analytical modelling and simulation [36]. In particular, for this work, to address the research questions, we created the artefacts Performance Assessment and Influence Mechanism and the PAAM, detailed in the following sections.

### 2.4. Addictions to Knowledge Base

As a result of DSR, the research activity and experience gained from field testing the artefacts in the environment, contribute to the knowledge base of scientific foundations. These contributions are key selling points to the academic and practitioner audience [38,39]. Thus, the contribution of the present work is a set of performance indicators to evaluate collaboration in a CBE and a simulation model, the PAAM, to assess and understand how the performance measures can influence the organisations to contribute to better CBE performance and sustainability.

## 3. Performance Assessment and Adjustment Model (PAAM)

The PAAM includes a Performance Assessment for CBEs, which is composed of the performance indicators briefly described in Table 3 and allows the CBE collaboration performance to be assessed through the indicators adopted by the CBE Manager: CI measures the contribution related to the new collaboration opportunities created and accepted in the CBE, PI measures the prominence among organisations related to their involvement in collaboration, and the II measures the creation of new products, services, or patents correlated with collaboration.

Each performance indicator has a weight that the CBE Manager can change with the purpose of influencing the organisations' behaviour to perform better in the desired direction: to give more collaboration contribution (increasing the *contact rate*, related to CI), to have more prominence (increasing the *accept rate*, related to PI), or to be more innovative (increasing the *new products rate*, related to II). These are the foundations of the Influence Mechanism, also included in the PAAM, summarised in Table 4. It is assumed that to influence the organisations' profile by a given factor of influence ($0 < FI < 1$), they react differently according to the weight of the respective indicator, meaning an increment of their *contact rate*, *accept rate* or *new products rate*, according to the formulas of Table 4. The formulas also consider an additional factor ($\pm F_e$) that allows introducing a random positive or negative influence due to exogenous causes.

**Table 4.** Summary of the Influence Mechanism, showing how to calculate a factor of influence (FI) in the profile of organisations.

| P. Ind. (Weight) | Influencing | Profile Affected by the FI |
|---|---|---|
| CI (*wCI*) | Contact rate | $Contact_{rate}+ = Contact_{rate} * wCI * \frac{FI}{wCI+wPI+wII} \pm F_e$ |
| PI (*wPI*) | Accept rate | $Accept_{rate}+ = Accept_{rate} * wPI * \frac{FI}{wCI+wPI+wII} \pm F_e$ |
| II (*wII*) | New prods. rate | $New\ prods_{rate}+ = New\ prods_{rate} * wII * \frac{FI}{wCI+wPI+wII} \pm F_e$ |

For the implementation of PAAM, we used AnyLogic tools [60] with simulation elements such as agent-based modelling (ABM) and system dynamics (SD). We used ABM

for simulating the actions and interactions of the autonomous agents (representing the organisations) in the environment (the CBE) and SD to provide stocks and flows to manage the organisations' resources. We also used discrete elements (statecharts, events, and timers) to combine the different techniques and models, to control state transitions, periods, or even capture exogenous values.

For the experimental evaluation, we used actual data collected during 2019 from three organisations operating in the IT sector in the same business ecosystem to create more realistic scenarios, such as the one shown in Figure 4. The collected data allowed us to define three classes of collaboration willingness, Class A, Class B, and Class C, shaped according to the framework of Table 1. As a result, the three profiles described in Table 5 establish three different behaviours of the organisations (the agents in the simulation model).

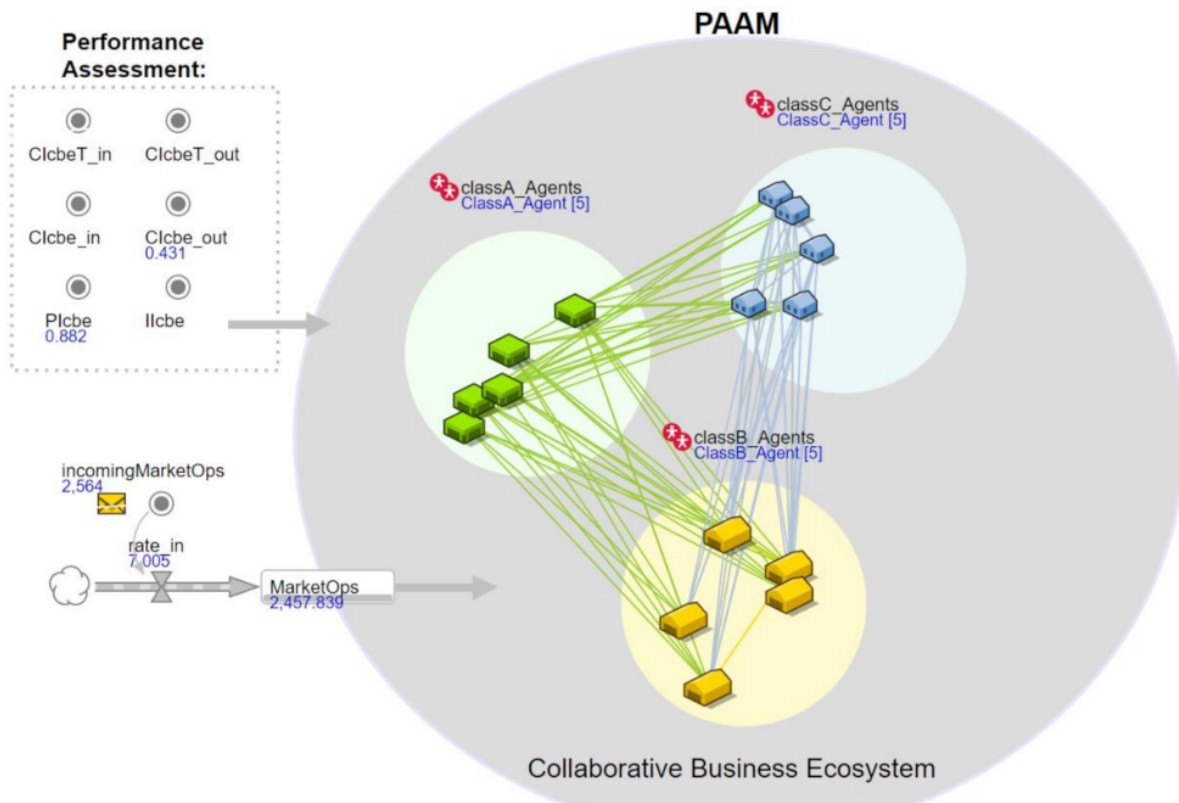

**Figure 4.** A view of the PAAM for a CBE instantiated with five organisations of each of the three different classes of collaboration willingness.

The total actual data collected of the three different organisations' profiles, required to shape all the parameters of the simulation model, are summarised in Table 6, containing:

- The estimated rate of the classes of collaboration willingness (*contact rate*, *accept rate* and *new products rate*);
- The human resources expressed in total persons and person-day;
- The percentage of human resources allocated by core activities (consulting, research and development, and inner tasks);
- The number of successful market opportunities calculated in person-days (interval between min and max, and mode, i.e., the typical duration);
- The percentage of the market opportunity (interval between min and max), i.e., the business units to distribute in the collaboration opportunities.

**Table 5.** Profile of the organisations in the CBE, classified into three classes of collaboration willingness (Class A, B, and C).

| Profile of Organisations in a CBE | | | | | | | | |
|---|---|---|---|---|---|---|---|---|
| Contact Rate | Accept Rate | New Products Rate | Contact Rate | Accept Rate | New Products Rate | Contact Rate | Accept Rate | New Products Rate |
| 0,56 | 0,00 | 0,06 | 0,06 | 1,00 | 0,13 | 0,60 | 0,65 | 0,63 |

**Table 6.** Summary of actual data collected from organisations of Class A, B, and C profiles.

| Organisations | Class A | | Class B | | Class C | |
|---|---|---|---|---|---|---|
| **Classes of Collaboration Willingness** | | | | | | |
| Contact rate | 0,56 | | 0,06 | | 0,60 | |
| Accept rate | 0,00 | | 1,00 | | 0,65 | |
| New products rate | 0,06 | | 0,13 | | 0,63 | |
| **Resources in Persons** | | | | | | |
| Total (persons) | 62 | | 16 | | 33 | |
| Total (person-day) | 13640 | | 3520 | | 7260 | |
| R&D | 2% | | 0% | | 6% | |
| Consulting | 74% | | 87% | | 85% | |
| Inner Tasks | 24% | | 13% | | 9% | |
| **Market Opportunities** | | | | | | |
| Duration | **min** | | **mode** | | **max** | |
| (person-day) | 0 | | 20 | | 100 | |
| **Business Units to Distribute** | | | | | | |
| Percentage of the market | **min** | **max** | **min** | **max** | **min** | **max** |
| opportunity | 4,0% | 7,2% | 0% | 16,7% | 0,4% | 4,5% |

After the PAAM was shaped with all data from Table 6, creating five agents of each profile (in the exemplified scenario), the model was run using a time window of one year (virtual time). A Poisson's distribution [60] generated 2400 market opportunities (two thousand plus 20% of opportunities with innovation). The results are represented in Figure 4, showing the connections (the collaboration opportunities) shared by the agents (the organisations) in the simulation environment (the CBE), which allowed us to calculate the adopted performance indicators ($CI_{out}$, and PI). These values are shown and discussed further in Section 5.

## 4. Model of the Agents

The organisations figure in the PAAM as agents, using ABM to model their collaboration behaviour in the CBE. Figure 5 shows a zoom-in of a Class A agent. A statechart controls the agent's state from *WaitNewOpportunity* to the state *InviteToCollaborate* when a new market opportunity is received. Then, the agent transits to a new state where he can decide (based on his profile) to invite another organisation to collaborate, creating a *CoOp* and forming a virtual organization with partners. On the other hand, when the recipient agent receives a new *CoOp*, he transits to the state *InviteReceived* and can or can not accept the collaboration based on his profile and available resources. Stocks from system

dynamics (SD) manage the agents' resources, and distribution functions [60] simulate the agents' decision to invite or accept *CoOps*, resulting in Formulas (1)–(3). The organisation's profile (*contact rate*, *accept rate* and *new products rate*) parameterizes a Bernoulli distribution, meaning that a higher parameter value results in a higher probability of having positive outcomes. Formula (4) uses a Triangular distribution to simulate the number of business units given to the other organisations resulting from the collaboration.

$$invite_{toCollaborate} = bernoulli(contactRate) \tag{1}$$

$$accept_{collaboration} = bornoulli(acceptRate) \tag{2}$$

$$accept_{collaboration} = bernoulli(newProductsRate) \tag{3}$$

$$businessUnits_{toDistribute} = triangular(minUnits, maxUnits) \tag{4}$$

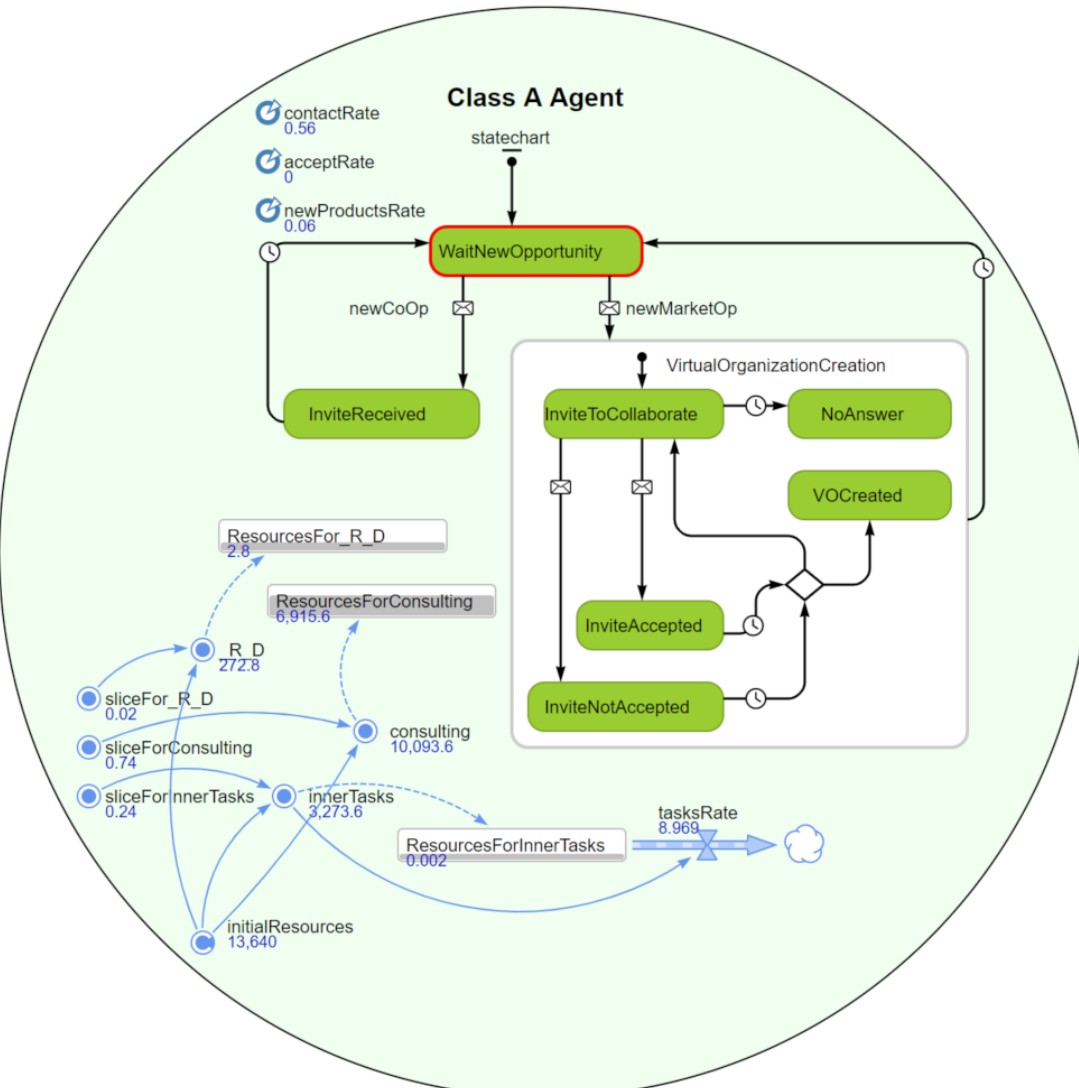

**Figure 5.** Zoom in the behaviour of a Class A organisation.

## 5. Simulation Results

The simulation results of the PAAM shown and discussed in this section corresponds to the scenario described in section three. The model was shaped with the actual data of Table 6, considering five organisations of each of the three profiles, running in time windows of one year of virtual time.

The presented experiment analyses the $CI_{out}$ and PI outcomes in three variants of the considered scenario: (a) measures without the influence of the performance indicators; (b) measures with a factor of influence $FI = 0.4$ and with performance indicators' weights of $wCI = 2$, $wPI = 1$, and $wII = 1$; and (c) measures with the same factor of influence but varying the perform indicators' weights to $wCI = 4$, $wPI = 1$, and $wII = 1$. The results are shown in a tabular and graphical representation using the Gephi tool [61].

Analysing the results of $CI_{out}$ (related to collaboration activity in the CBE) in the three simulation's scenarios (Table 7 and Figure 6), we can observe that almost all the organisations in scenario of Figure 6c tried to create more *CoOps* than in scenario of Figure 6b, because the influence mechanism increased the weight of $wCI$ from $wCI = 2$ to $wCI = 4$. More collaboration of all organisations in scenario of Figure 6c than in Figure 6b resulted in a lower value of $CI_{CBE}out$, meaning a more uniform collaboration. This trend is desirable, as it means more cohesion amongst all the CBE's organisations, thus contributing to their sustainability.

**Table 7.** $CI_{out}$ normalised measures for organisations and the CBE, using three scenarios of simulation: (a) without the influence mechanism; (b) after the influence considering $wCI = 2$; and (c) after increasing the weight to $wCI = 4$.

| Profile | $O_i$ | $CI_i$ out(a) | $CI_i$ out(b) | $CI_i$ out(c) |
|---|---|---|---|---|
| Organisations of Class A | 0 | 0,76 | 0,51 | 0,83 |
| | 1 | 0,69 | 0,59 | 0,53 |
| | 2 | 0,90 | 0,79 | 0,75 |
| | 3 | 0,83 | 0,72 | 0,97 |
| | 4 | 0,93 | 1,00 | 0,89 |
| Organisations of Class B | 5 | 0,03 | 0,03 | 0,06 |
| | 6 | 0,14 | 0,08 | 0,08 |
| | 7 | 0,10 | 0,05 | 0,14 |
| | 8 | 0,03 | 0,00 | 0,06 |
| | 9 | 0,00 | 0,05 | 0,14 |
| Organisations of Class C | 10 | 0,83 | 0,74 | 0,94 |
| | 11 | 1,00 | 0,69 | 0,86 |
| | 12 | 1,00 | 0,77 | 0,78 |
| | 13 | 0,79 | 0,79 | 0,78 |
| | 14 | 0,93 | 0,74 | 1,00 |
| **$CI_{CBE}out$** | | **0,43** | **0,53** | **0,44** |

Analysing the results of PI (related to prominence) in the three simulations' scenarios (Table 8 and Figure 7), we can observe that, because the profile of Class A organisations has an *Accept rate* = 0, the PI indicator equals zero and remains zero after the influence mechanism. For the other classes, we can perceive that, in scenarios of Figure 7b,c, the organisations did not change their behaviour significantly to acquire more prestige due to the low value of the weight $wPI = 1$. The CBE remains polarized in just a few organisations that come out with a more significant predominance. However, due to the increase in $wCI$ weight, from $wCI = 2$ to $wCI = 4$, inducing more collaboration created in the CBE resulted in lower values of the $PI_{CBE}$ in scenarios of Figure 7b,c. These lower values improve the CBE as the more uniform distribution of the organisations' prestige reduces the concentration of power in just a few organisations, thus contributing to the CBE sustainability.

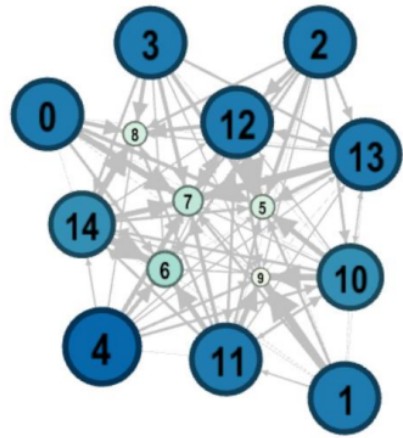

(**a**) CI$_{out}$ results before the influence.

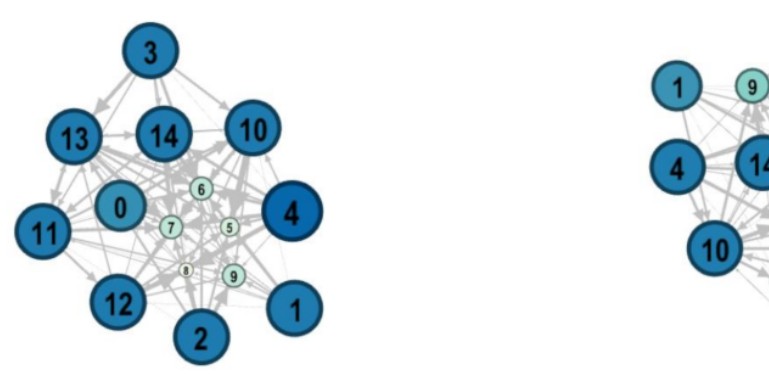

(**b**) CI$_{out}$ measures after the influence.     (**c**) CI$_{out}$ measures after the influence varying the indicators' weights.

**Figure 6.** CIout measures before the CBE's influence; the nodes' sizes are related to the CI value, i.e., the bigger the nodes, the greater the indicators' values; the connections' strengths are weighted by the number of CoOps exchanged.

**Table 8.** PI normalised measures for organisations and the CBE, using three scenarios of simulation: (a) without the influence mechanism; (b) after the influence considering *wCI* = 2; and (c) after increasing the weight to *wCI* = 4.

| Profile | O$_i$ | PI$_i$ (a) | PI$_i$ (b) | PI$_i$ (c) |
|---|---|---|---|---|
| Organisations of Class A | 0 | 0,00 | 0,00 | 0,00 |
| | 1 | 0,00 | 0,00 | 0,00 |
| | 2 | 0,00 | 0,00 | 0,00 |
| | 3 | 0,00 | 0,00 | 0,00 |
| | 4 | 0,00 | 0,00 | 0,00 |
| Organisations of Class B | 5 | 0,00 | 0,00 | 0,00 |
| | 6 | 0,00 | 0,15 | 0,00 |
| | 7 | 0,13 | 0,20 | 0,04 |
| | 8 | 0,00 | 0,00 | 0,00 |
| | 9 | 0,00 | 0,05 | 0,15 |
| Organisations of Class C | 10 | 0,06 | 0,21 | 0,60 |
| | 11 | 0,32 | 0,62 | 1,00 |
| | 12 | 1,00 | 0,41 | 0,71 |
| | 13 | 0,08 | 1,00 | 0,12 |
| | 14 | 0,18 | 0,06 | 0,46 |
| **PI$_{CBE}$** | | **0,88** | **0,82** | **0,79** |

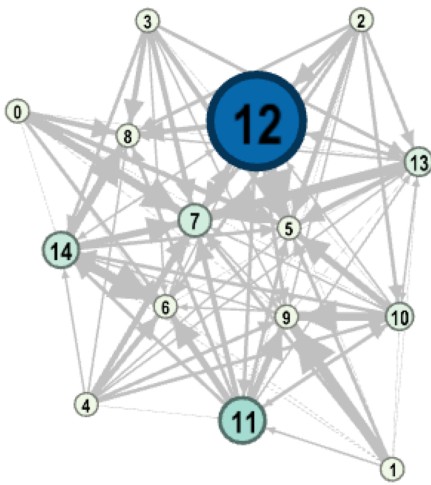

(**a**) PI results before the influence.

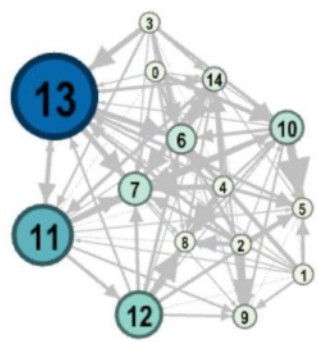

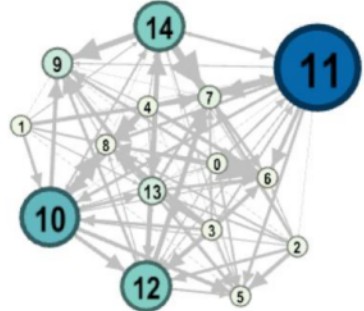

(**b**) PI measures after the influence.  (**c**) PI measures after the influence varying the indicators' weights.

**Figure 7.** PI measures before the CBE's influence; the nodes' sizes are related to the PI value, i.e., the bigger the nodes, the greater the indicators' values; the connections' strengths are weighted by the number of *CoOps* exchanged.

The results respond to the research questions:

Response to **RQ1.** *The adopted performance indicators can assess the collaboration of the organisations in the CBE, measuring their popularity related to the* $CI_{in}$, *activity related to* $CI_{out}$, *and prominence related to PI. The CBE as a whole can also be assessed by measuring the average contribution by an organisation and the uniformity of collaboration among organisations. A more uniform collaboration means a more sustainable CBE, as it means a more cohesive group.*

Response to **RQ2.** *The CBE manager can adopt a set of performance indicators to influence the behaviour of the organisations and vary the weights to induce responses in the desired direction of more sustainability.*

## 6. Conclusions

The experimental evaluation results using the PAAM shaped with actual data collected from organisations running in the same business ecosystem showed that the research questions had been answered. The simulation outcomes of the analysed indicators, $CI_{out}$ and PI, showed that the collaboration of the organisations in a CBE can be measured and influenced, causing an adjustment on their behaviour to improve the performance and sustainability of the ecosystem.

The contribution of this work is a set of performance indicators suitable for CBEs and a simulation model, the PAAM, that the CBE Manager can use to explore simulation scenarios in the search for a better balance leading to improved performance in the ecosystem due to the benefits that collaboration can bring [12,40–42], thus contributing to an improvement in collaboration sustainability [41].

For future work, various other simulation scenarios should be analysed adopting different sets of the proposed performance indicators ($CI_{in}$, $CI_{out}$, PI, and II), helping to understand the dynamics of a CBE to improve the simulation model and the influence mechanism. The main limitation of this work is that the actual data used to shape the simulation model came from the IT services industry and was extrapolated to represent fifteen organisations in the same CBE. This context may not reflect the reality of other business ecosystems.

**Author Contributions:** The presented work is part of the research made by the author P.G. to achieve a PhD with the dissertation Performance Indicators for Collaborative Business Ecosystems. The corresponding author, professor L.M.C.-M. is the supervisor. All authors have read and agreed to the published version of the manuscript.

**Funding:** Fundação para a Ciência e Tecnologia, project UIDB/00066/2020.

**Institutional Review Board Statement:** Not applicable.

**Informed Consent Statement:** Not applicable.

**Data Availability Statement:** Data is contained within the article.

**Acknowledgments:** This work benefited from the ongoing research within the CoDIS (Collaborative Networks and Distributed Industrial Systems Group), which is part of both the Nova University of Lisbon (UNL)—School of Science and Technology and the UNINOVA—CTS (Center of Technology and Systems). Partial support also comes from Fundação para a Ciência e Tecnologia through the program UIDB/00066/2020.

**Conflicts of Interest:** The authors declare no conflict of interest.

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
