# Peer review of "Assessment of Sustainable Collaboration in Collaborative Business Ecosystems†"

_computers, doi:10.3390/computers10120167_

Round 1
Reviewer 1 Report
Despite the relevance of the research and the rather high level of scientific character of the research, the choice of the instrumentation seems to be controversial. In the author's formulation of the problem, rather than networks, but multiplex networks are the basis for interorganizational interactions. Instead of models for collaborative networked, it is necessary to use the multiplex models toolkit, well known and adapted over the past 10-15 years. In addition, the choice of network parameters (table 2) and indicators (table 3) is not reasoned. It is not clear from the text of the article how the network parameters were calculated. The authors should also expand and substantiate the conclusions following from the work
Reviewer 2 Report
Well-done research. The results deserve publication. Some errors in the text and formulas should be eliminated.
Reviewer 3 Report
The paper contains minor shortcomings. In order to improve, the following should be considered:
- Figure 4 and table 5 require detailed explanations.
- The authors should clearly answer to research question no.1. The authors should clearly highlight the research results.
- The authors should clearly answer to research question no.2. The authors should clearly highlight the research results.
- The authors should improve the conclusion. The conclusion must give clear answers on: what the problem was, how was it solved, what are the results/solutions, what is solved.
Round 2
Reviewer 1 Report
The authors left without comment all my previous comments. Therefore, I cannot support this work.